# Endogenous siRNAs promote proteostasis and longevity in germline-less *Caenorhabditis elegans*

Moran Cohen-Berkman[1], Reut Dudkevich[1], Shani Ben-Hamo[1], Alla Fishman[2], Yehuda Salzberg[3], Hiba Waldman Ben-Asher[1], Ayelet T Lamm[2], Sivan Henis-Korenblit[1]*

[1]The Mina & Everard Goodman Faculty of Life Sciences, Bar-Ilan University, Ramat-Gan, Israel; [2]Faculty of Biology, Technion-Israel Institute of Technology, Technion City, Haifa, Israel; [3]Department of Neurobiology, Weizmann Institute of Science, Rehovot, Israel

**Abstract** How lifespan and the rate of aging are set is a key problem in biology. Small RNAs are conserved molecules that impact diverse biological processes through the control of gene expression. However, in contrast to miRNAs, the role of endo-siRNAs in aging remains unexplored. Here, by combining deep sequencing and genomic and genetic approaches in *Caenorhabditis elegans*, we reveal an unprecedented role for endo-siRNA molecules in the maintenance of proteostasis and lifespan extension in germline-less animals. Furthermore, we identify an endo-siRNA-regulated tyrosine phosphatase, which limits the longevity of germline-less animals by restricting the activity of the heat shock transcription factor HSF-1. Altogether, our findings point to endo-siRNAs as a link between germline removal and the HSF-1 proteostasis and longevity-promoting somatic pathway. This establishes a role for endo siRNAs in the aging process and identifies downstream genes and physiological processes that are regulated by the endo siRNAs to affect longevity.

*For correspondence:
sivan.korenblit@biu.ac.il

Competing interests: The authors declare that no competing interests exist.

## Introduction

Aging is a major risk factor for chronic age-related diseases, which have become a major cause of death in the elderly (*Matus et al., 2011*). Such pathologies and aging share a common set of basic biological mechanisms, including a failure to maintain the homeostasis of the proteome (proteostasis) with age. Given the high conservation of the aging and proteostasis-promoting pathways between low and high organisms, analysis of these pathways in *Caenorhabditis elegans* has proven to be valuable for the understanding of aging and proteostasis in all animals, including mammals.

Dedicated signaling pathways, which coordinate cellular processes that maintain protein homeostasis, have evolved to prevent the grave consequences associated with the accumulation of misfolded proteins (*Taylor et al., 2014*). These pathways are triggered by the accumulation of misfolded proteins in different cell compartments and initiate processes that maintain a functional protein-folding environment by controlling translation rate, increasing expression of chaperones, and enhancing the protein degradation machinery.

In the cytoplasm, the proteostasis stress response pathway is governed by the transcription factor HSF1 (*Akerfelt et al., 2010*). In *C. elegans*, HSF-1 overexpression is sufficient for extending lifespan and is important for lifespan extension by most longevity pathways (*Hsu et al., 2003*; *Baird et al., 2014*; *Morley and Morimoto, 2004*). Although the proteostasis stress response pathways assure proteome homeostasis in young animals during development, they lose responsiveness and fail to protect the proteome of aging animals (*Shemesh et al., 2013*; *Labbadia and Morimoto, 2015*;

*Taylor and Dillin, 2011*). This may increase the risk for protein conformational diseases, such as Alzheimer's disease, Huntington's disease, Parkinson's disease *etc.* (*Matus et al., 2011*; *López-Otín et al., 2013*).

Consistent with the importance of proteostasis maintenance for proper health and function, mutations and treatments that extend lifespan and healthspan maintain proteostasis in aging animals. Improved proteostasis can be achieved by boosting protein degradation pathways in the soma (*Safra et al., 2014*; *Henis-Korenblit et al., 2010*; *Vilchez et al., 2012*), or by postponing the age-dependent decline in the responsiveness of the stress responses (*Shemesh et al., 2013*; *Labbadia and Morimoto, 2015*).

In *C. elegans* (*Hsin and Kenyon, 1999*) and *Drosophila* (*Flatt and Schmidt, 2009*), germline depletion extends lifespan. Likewise, lifespan extension can result from ovarian transplantation experiments in mice (*Mason et al., 2009*) and castration in men (*Min et al., 2012*). These suggest that reproductive control on lifespan might be conserved in mammals as well. In addition to extending lifespan, germline depletion also promotes proteostasis. The improved proteostasis of germline-less *C. elegans* is achieved by reducing the repressive chromatin marks at HSF1-regulated stress-responsive genes. In turn, the removal of repressive chromatin marks delays the age-dependent collapse of the proteostasis promoting pathways (*Shemesh et al., 2013*; *Labbadia and Morimoto, 2015*).

Studies of the last decade identified about a dozen genes that function in the *C. elegans* reproductive-longevity pathway. Many of these genes encode or regulate transcription factors, which are activated in the intestine upon germline removal. Little is known about how depletion of germline stem cells regulates these transcription factors, with the exception of DAF-16 (*Antebi, 2013*). The germline-regulated transcription factors remodel the transcriptional landscape in germline-less animals. Germline-regulated genes are enriched in proteostasis, innate immunity, and metabolism-related genes, altering the physiology of the animals and promoting longevity (*McCormick et al., 2012*). Accordingly, germline depletion enhances oxidative stress resistance and immunity (*Alper et al., 2010*; *Libina et al., 2003*), modulates fat metabolism (*Wang et al., 2008*; *Ratnappan et al., 2014*; *Steinbaugh et al., 2015*), induces autophagy (*Lapierre et al., 2013*), and boosts proteostasis-related stress responses in aging animals (*Shemesh et al., 2013*).

Small RNAs and their Argonaute cofactors are conserved components of eukaryotic organisms. Along with transcription factors and transcription regulators, the small RNA silencing pathways impose a layer of gene regulation, which affects diverse biological processes. This is achieved by the generation of short antisense RNAs that act in the cytoplasm, where they interfere with gene expression by inhibiting translation, by degrading cytoplasmic mRNA, or by altering mRNA storage (*Grishok, 2013*). Short antisense RNAs also target chromatin modifications in the nucleus, generating epigenetic changes (*Burton et al., 2011*). In *C. elegans*, small RNAs can move between tissues (*Winston et al., 2002*) and be passed along several generations (*Rechavi and Lev, 2017*).

The three main endogenous small RNA pathways in *C. elegans* include miRNAs, endogenous small interfering RNAs (endo-siRNAs), and PIWI (P-element-induced wimpy testis) interacting RNAs (piRNAs). Each of these pathways uses RNAs with different characteristics and involves both distinct and overlapping enzymes and Argonautes. Although small noncoding RNAs impact many biological processes, in the context of aging, studies mainly focused on miRNAs. Multiple studies followed the age-associated changes in expression of miRNAs in *C. elegans* (*Kato et al., 2011*; *de Lencastre et al., 2010*; *Aalto et al., 2018*; *Ibáñez-Ventoso, 2006*). Furthermore, life-extending and life-shortening properties have been attributed to specific miRNAs (*Shen et al., 2012*; *Boulias and Horvitz, 2012*; *Boehm and Slack, 2005*) and miRNA-dedicated Argonautes (*Aalto et al., 2018*). These include several miRNAs that facilitate the localization and transcriptional activity of DAF-16 in the intestine of germline-less animals (*Shen et al., 2012*; *Boulias and Horvitz, 2012*).

In contrast to miRNAs, much less is known about the physiological roles of naturally-produced endogenous siRNAs that align and complement multiple coding and non-coding loci across the genome (*Blumenfeld and Jose, 2016*; *Gu et al., 2009*). Thus far, endo-siRNAs in *C. elegans* have been primarily implicated in immune surveillance (*Fischer, 2010*; *Rechavi et al., 2011*) and the transfer of stress resistance between generations (*Rechavi et al., 2014*; *Kishimoto et al., 2017*). Nevertheless, the functional role of endo siRNAs in the regulation of aging remains largely unexplored. This is in spite of the fact that a study of global small RNA profiling over the course of *C. elegans* aging identified an age-dependent increase in the expression of different endo-siRNA (*Kato et al.,*

*2011*). Furthermore, another study reported that endo-siRNAs regulate the lifespan of the fly (*Lim et al., 2011*). Here, by combining deep sequencing and genomic and genetic approaches in *C. elegans*, we have established a role of endo-siRNAs in lifespan extension and the regulation of the proteostasis-promoting transcription factor HSF-1 in germline-less animals and have also identified direct and indirect aging-related targets of this silencing pathway.

## Results

### Endo-siRNAs contribute to the longevity of germline-less animals

To examine if endo-siRNAs are implicated in the longevity of germline-less animals, we made use of the well-characterized *dcr-1(mg375) C. elegans* mutant strain (*Welker et al., 2010*). Dicer is a member of the RNase III family of nucleases that degrade double-stranded RNA (dsRNA). Dicer processes exogenous dsRNA as well as endogenous dsRNA of which miRNAs and endo-siRNAs are produced. Unlike most *dcr-1* alleles that interfere with the processing of a variety of small RNAs, *dcr-1(mg375)* mutants have a point mutation in the helicase domain of the dicer enzyme. This point mutation disrupts the processing of a subset of endo-siRNAs, without affecting the processing of other small RNA molecules (*Welker et al., 2010*). To limit germline proliferation, we made use of *glp-1(e2144)* mutants, which carry a temperature-sensitive notch receptor required for germline stem cells (GSC) proliferation (*Priess et al., 1987*). For simplicity, we will refer henceforth to germline-less *glp-1* animals raised at the restrictive temperature as GSC(-) animals.

First, we generated *dcr-1(mg375) glp-1* double mutants and followed their lifespan when raised from eggs to adulthood at restrictive temperature. As expected, *glp-1* GSC(-) animals exhibited extended lifespan compared to wild-type animals (*Figure 1A*). However, we found that the *dcr-1 (mg375)* mutation shortened the lifespan of GSC(-) animals to a greater extent than in animals with an intact germline (*Figure 1A* and *Supplementary file 1*). Similarly, limiting germline expansion by dietary supplementation of dihomo-γ-linolenic acid (DGLA) (*Watts and Browse, 2006*) extended the lifespan of wild-type animals (*Shemesh et al., 2017*; *O'Rourke et al., 2013*), but failed to extend the lifespan of *dcr-1(mg375)* mutants (*Figure 1B* and *Supplementary file 1*). Together, these findings imply that endo-siRNAs that depend on the helicase activity of dicer may be implicated in longevity induced by germline removal.

We then examined if mutations in additional components, which are specifically required for the processing of endo-siRNA but not directly implicated in the processing of other small RNAs, affected the lifespan of GSC(-) animals. To this end, we generated *glp-1* double mutants with mutations in the *rrf-3*, *ergo-1*, and *nrde-3* genes. *rrf-3* encodes an RNA-directed RNA polymerase (RdRP), which uses single-stranded RNA as template for second-strand synthesis (*Gent et al., 2010*). *ergo-1* encodes an endo-siRNA-specific Argonaute, which stabilizes the initial class of 26G endo-siRNAs (*Vasale et al., 2010*). NRDE-3 is an Argonaute that functions in nuclear RNA interference (RNAi) (*Guang et al., 2008*). In all cases, the lifespan extension conferred by germline removal was significantly curtailed by mutations that perturbed different steps in the processing of endo-siRNAs, with less of an effect on the lifespan of animals with an intact reproductive system (*Figure 1C–E* and *Supplementary file 1*). The finding of similar shortening of the lifespan of GSC(-) animals by multiple mutations that affect endo-siRNA processing strongly implicates endo-siRNAs in the lifespan extension of GSC(-) animals. Furthermore, this reduces the likelihood that the observed differences in lifespan are due to background mutations in non-outcrossed strains *dcr-1(mg375)* and *nrde-3(gg66)*. Thus, we conclude that the processing of a subset of siRNAs, whose processing is *dcr-1* helicase, *rrf-3*, *ergo-1*, and *nrde-3*-dependent, contributes to the lifespan extension of GSC(-) animals.

In *C. elegans*, small RNA molecules spread between cells via SID-1 dsRNA channels (*Shih and Hunter, 2011*). To explore if endo-siRNAs act in a hormonal-like fashion to promote longevity, we examined whether the longevity of GSC(-) animals is dependent upon SID-1 channels. We found that GSC(-) *glp-1; sid-1* double mutants were long-lived similarly to GSC(-) *glp-1* single mutants (*Figure 1F* and *Supplementary file 1*). Thus, there was no need for SID-1-dependent uptake of the endo-siRNA molecules by neighboring cells for the longevity of *glp-1* mutants. This could be either because the silencing takes place in the same cells that produce the lifespan-regulatory small RNA molecules, or because an alternative RNA channel mediates the spread of the endo-siRNAs between tissues.

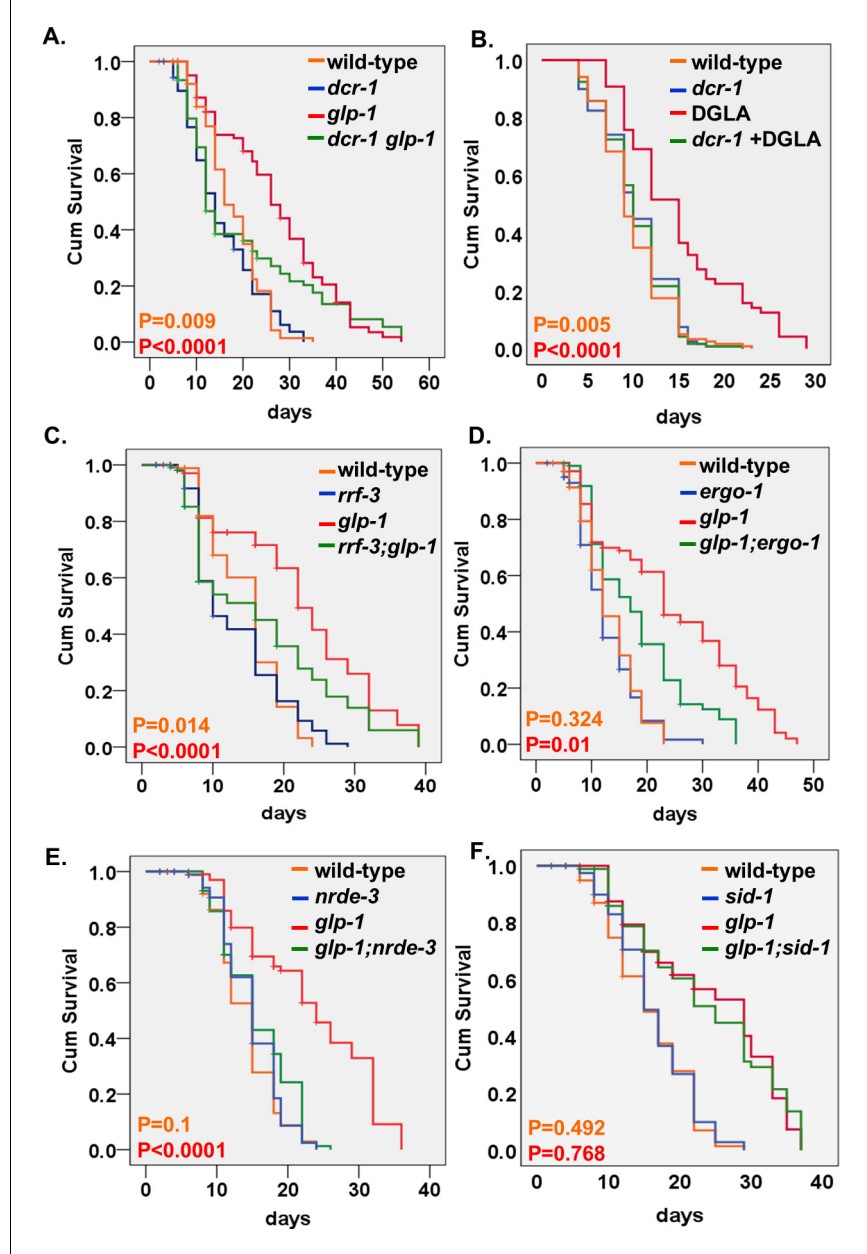

**Figure 1.** Endo-siRNAs are required for the longevity of GSC(-) animals. (A,C–E) Impairment of the endo-siRNA pathway by *dcr-1(mg375)*, *rrf-3(pk1426)*, *ergo-1(gg98)*, or *nrde-3(gg66)* compromises the longevity of *glp-1* mutants. (B) The longevity conferred by germline depletion via DGLA supplementation is perturbed by the *mg375* mutation in the *dcr-1* helicase domain. (F) Impairment of the dsRNA channel *sid-1(pk3321)* does not compromise the longevity of *glp-1* mutants. Breslow (Generalized Wilcoxon) P-values between endo-siRNA mutants and corresponding animals with intact endo-siRNA are indicated between GSC(+) animals (in orange) or between GSC (-) animals (in red). See *Supplementary file 1*. Note that the *dcr-1* and *nrde-3* mutants have not been outcrossed. This may affect their lifespan phenotypes.

## Endo-siRNAs promote chaperone expression in GSC(-) animals

Germ cell depletion results in significant remodeling of the animal's transcriptome, promoting the expression of genes that drive proteostasis, autophagy, innate immunity, lipid metabolism and more (*Antebi, 2013*). In order to identify which downstream physiological processes are hindered by the depletion of endo-siRNA in GSC(-) animals, we compared the mRNA transcriptomes of GSC(-) animals and wild-type animals upon interference with the processing of endo-siRNA. Even though

mutations in several endo-siRNA related genes affected the lifespan of GSC(-) animals, we performed this analysis using the *dcr-1(mg375)* point mutation, due to its relative focused effect on only a subset of endo-siRNAs molecules (*Welker et al., 2010*).

We first focused on a group of 72 genes whose mRNA levels consistently decreased by more than 1.5 fold (p-value<0.05) in *dcr-1 glp-1* double mutants compared to *glp-1* single mutants (*Supplementary file 2*). Since siRNAs downregulate gene expression (i.e., lack of siRNA results in upregulation of their target genes), these genes cannot be direct targets of the siRNA pathway. However, their altered expression upon siRNA inactivation may point out downstream processes affected by the siRNA activity. A protein-protein interaction network of these 72 genes using STRING (*Szklarczyk et al., 2019*) highlighted a group of 10 interacting genes, composed mostly of chaperone-encoding genes (*Figure 2—figure supplement 1*). GO enrichment analysis (*Ashburner et al., 2000*) of the 72 genes identified enrichment in genes related to cellular response to unfolded proteins as relatively down-regulated in the *dcr-1 glp-1* double mutants (p-value<$1*10^{-4}$, *Supplementary file 3*). Interestingly, out of the group of 72 genes, 15 genes are known to be upregulated in an *hsf-1*-dependent manner upon heat shock (*Brunquell et al., 2016*). This is a significant enrichment in the amount of HSF-1-regulated genes, more than expected by chance (chi-square with yates correction p-value<$1*10^{-4}$). We confirmed the downregulation of three of these genes in *dcr-1 glp-1* double mutants compared to *glp-1* mutants by qRT-PCR (*Figure 2A*). We hypothesize that the growth conditions of the animals at 25 degrees from eggs to adulthood, after prior cultivation in 20 degrees, resulted in a mild heat shock response, allowing the detection of heat-induced genes. Together, these results suggest that in the absence of some endo-siRNAs, expression of a set of cytosolic chaperones and additional HSF-1-regulated genes is compromised in GSC(-) animals.

## Endo-siRNAs are required for activation of the heat shock response in GSC(-) animals

Our data indicate that endo-siRNAs are important for the expression of a set of HSF-1 target genes in adult GSC(-) animals. Hence, we first checked if there was a decrease in HSF-1 protein levels in *dcr-1 glp-1* animals compared to *glp-1* animals. Analysis of the levels of endogenous HSF-1 protein in these mutants demonstrated that HSF-1 levels were higher in GSC(-) animals compared to wild-type animals (p=0.004, *Figure 2B*). Furthermore, HSF-1 levels were not reduced upon dicer inactivation in GSC(-) animals (in fact HSF-1 levels were higher in *dcr-1 glp-1* double mutants compared to *glp-1* single mutants in 5/7 experiments, p=0.054, *Figure 2B*). These results demonstrate that endo-siRNAs are not required for HSF-1 expression in GSC(-) animals.

We next checked whether the endo-siRNA pathway affected HSF-1 activation. One hallmark of HSF-1 activation upon heat-shock is its rapid redistribution into sub-nuclear structures, which share many properties with human nuclear stress granules (*Morton and Lamitina, 2013*). The formation of these foci is dependent upon the DNA binding domain of HSF-1 and they co-localize with markers of active transcription. We used the same single copy HSF-1::GFP translational fusion strain to follow its organization into foci in different genetic backgrounds. As reported (*Morton and Lamitina, 2013*), under non-stress conditions, HSF-1::GFP was found primarily in the nucleus but not in foci. Following a 10 min heat shock, day three wild-type animals had on average two foci per hypodermal cell. In long-lived *glp-1* animals, we observed increased levels of these foci under heat shock conditions (on average 3.6 foci per hypodermal cell). In contrast, GSC(-) animals impaired in their endo-siRNA pathway had only 1.4 foci on average per hypodermal cell (*Figure 2C–D*). These results demonstrate that *dcr-1*-dependent endo-siRNA molecules are required for HSF-1 foci formation upon heat shock in GSC(-) animals.

## Endo-siRNAs are required for proteostasis maintenance in GSC(-) animals

A low amount of chaperones and reduced ability of HSF-1 to form foci may render *dcr-1 glp-1* animals sensitive to proteostasis challenges. To test this, we exposed late day two animals to a prolonged heat shock and followed their survival after a recovery period. As reported (*Shemesh et al., 2013*; *Libina et al., 2003*), *glp-1* animals were more resistant to heat shock than were wild-type animals. While 71% of the *glp-1* animals survived the heat shock, only 38% of the wild-type animals

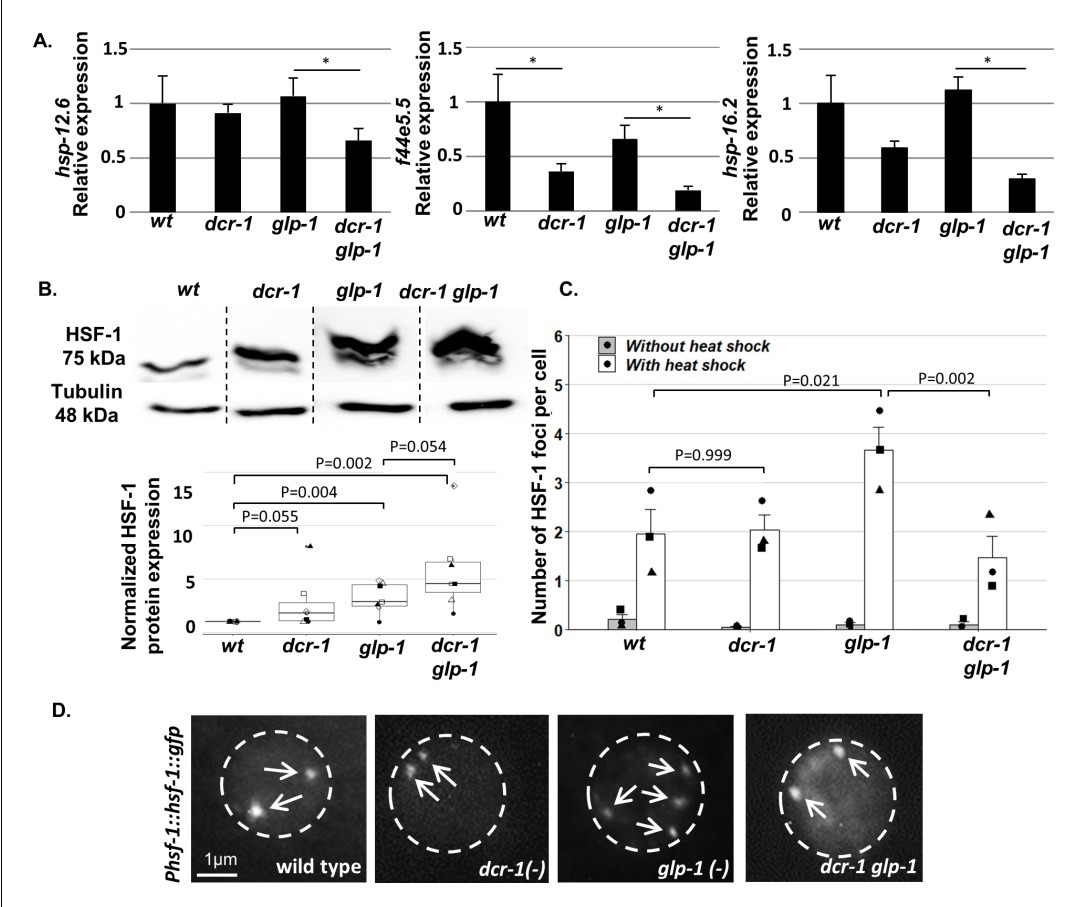

**Figure 2.** Endo-siRNAs are required for HSF-1 activation in GSC(-) animals. (A) qRT-PCR of the indicated genes on day 1 of adulthood. Asterisks mark Student's t-test values of p value<0.05 (N = 4). Note that the *dcr-1* mutation reduced the transcript levels of all three chaperone genes in GSC(-) animals, consistent with the possibility that the activity of their upstream transcription factor HSF-1 has been compromised. Interestingly, the *dcr-1* mutation also affected the levels of the *f44e5.5* transcript in GSC(+) animals. Nevertheless, it did not significantly affect the transcript levels of the *hsp-16.2* and *hsp-12.6* chaperones in GSC(+) animals. Given that HSF-1 and some of its targets are expressed also in the germline (*Ooi and Prahlad, 2017*), to avoid biases due to the presence/absence of the germline tissue, comparisons should be made within GSC(-) animals or within GSC(+) animals (*McCormick et al., 2012*; *Steinbaugh et al., 2015*). See also *Figure 2—figure supplement 1* and *Supplementary file 2* and *Supplementary file 3*. (B) Representative western blot of endogenous HSF-1 in day one animals (upper panel) compared to loading control (lower panel). Boxplots represent the distribution of normalized HSF-1 levels per strain. Different shapes represent independent experiments (N = 7). P-values of One-Sample Test and One-Way ANOVA followed by Tukey's post hoc analysis across all seven experiments are indicated. See *Supplementary file 8* for statistic details. (C) Bars represent mean of means of the number of HSF-1::GFP nuclear foci per hypodermal cell. Dots represent mean number of HSF-1 foci per cell with different shapes representing independent experiments. At least 140 cells per genotype were scored in a total of 3 independent experiments. P-values determined by One-Way ANOVA followed by Tukey's post hoc analysis are indicated. Data are presented as mean ± SEM. See *Supplementary file 8* for statistic details. (D) Representative fluorescent micrographs of hypodermal cell nuclei in day three adults, harboring a single copy of the *Phsf-1::hsf-1::gfp* transgene upon heat shock stress. Exposures and contrast were adjusted for each picture independently to best emphasize foci amount. Nucleus boundaries are circled.

The online version of this article includes the following figure supplement(s) for figure 2:

**Figure supplement 1.** String analysis of 72 genes whose levels decreased by more than 1.5 fold in *dcr-1 glp-1* double mutants compared to *glp-1* mutant.

survived the same stress (*Figure 3A,P*-value<0.001). Strikingly, the *dcr-1 glp-1* mutants were sensitive to the heat shock, similar to the wild-type animals, with only 39% survival (*Figure 3A,P*-value=0.99). Similar sensitivity to heat shock was observed upon perturbation of the endo-siRNA pathway in *glp-1* mutants by a mutation in the *rrf-3* gene (*Figure 3B*). The finding of similar sensitivity of GSC(-) animals to heat stress by two independent mutations that affect endo-siRNA processing strongly implicates endo-siRNAs in the heat shock resistance of GSC(-) animals.

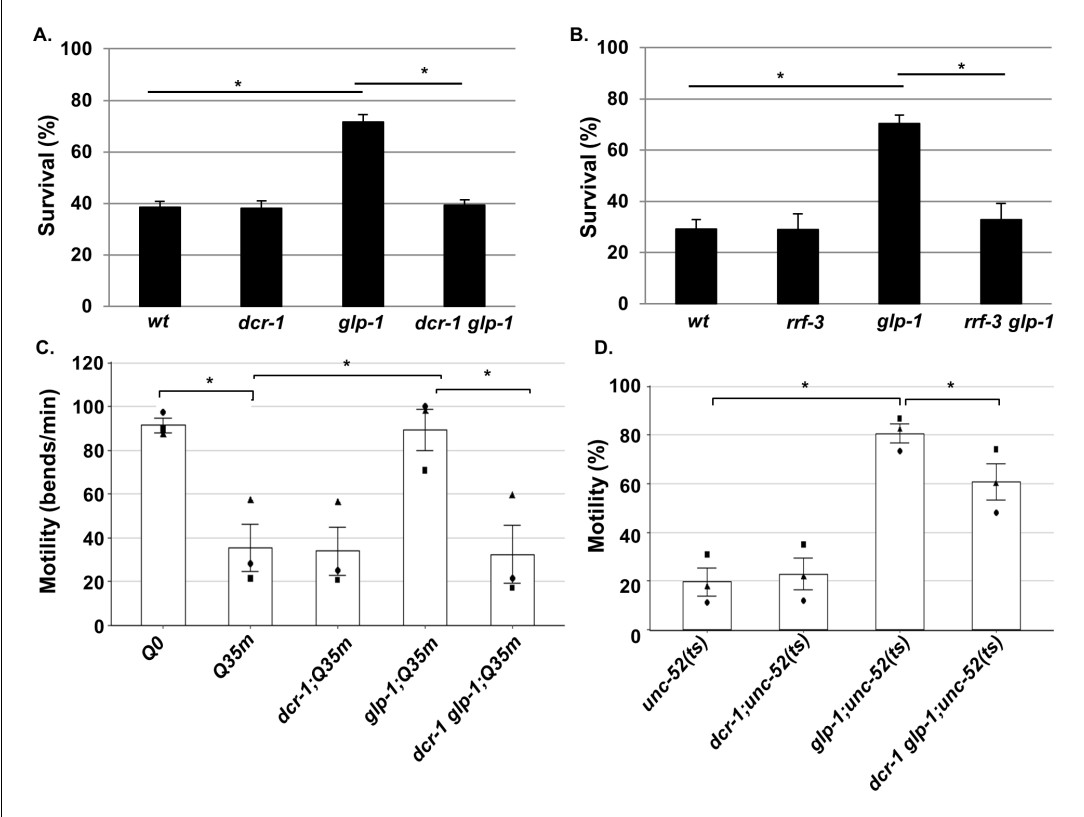

**Figure 3.** Endo-siRNAs are required for proteostasis maintenance in GSC(-) animals. (A–B) Thermo-resistance was examined in age-synchronized animals subjected to heat shock (37˚C, 9 hr) on day 2 of adulthood. Survival was assayed after 5 hr of recovery at 25˚C (120 animals per treatment, N = 3). Asterisks mark Cochran-Mantel-Haenszel Test values of p<0.001. (C) Bars represent mean of the percentage of motile animals scored in age-synchronized day 5 *Q35m* or *glp-1;Q35m* animals (more than 45 animals per treatment, N = 3). Different shapes represent mean motility in independent experiments. Asterisks mark p-values<0.001 determined by One-Way ANOVA followed by Tukey's post hoc analysis. (D) Bars represent mean of means of the number of body bends per minute in age-synchronized day four *unc-52(ts)* animals. Animals were raised at 25˚C till day 1 of adulthood, and shifted to the permissive temperature (15˚C) thereafter. A total of 180 animals per strain were scored in three independent experiments. Different shapes represent mean motility in independent experiments. Asterisks mark Cochran-Mantel-Haenszel test values of p<0.001. Data are presented as mean ± SEM. See *Supplementary file 8* for statistic details.

Furthermore, this reduces the likelihood that the observed differences in heat sensitivity are due to background mutations in the non-outcrossed *dcr-1(mg375)* strain.

Expression of toxic aggregating proteins such as Poly-Q rich proteins perturbs proteostasis. Specifically, the expression of these toxic proteins in the muscle cells causes the animals to undergo age-dependent loss of motility (*Morley et al., 2002*). Hence, we examined the motility of wild-type and GSC(-) animals, expressing toxic poly Q35 fused to YFP in their muscles in the presence or absence of an intact endo-siRNA pathway. The motility of the animals was determined by the number of swimming strokes they performed when placed in liquid on day 5 of adulthood. As reported (*Shemesh et al., 2013*; *Labbadia and Morimoto, 2015*), wild-type Q35 animals showed reduced motility compared to their GSC(-) Q35 counterparts (*Figure 3C*). Perturbation of the endo-siRNA pathway by a point mutation in the *dcr-1* gene reduced the motility of *glp-1* mutants to the level of wild-type animals (*Figure 3C*). Altogether, these experiments suggest that endo-siRNA molecules contribute to the superior proteostasis state of GSC(-) animals.

Finally, we also asked whether endo-siRNAs contributed to proper protein folding. For this purpose, we used the *unc-52(e669su250)* allele, harboring a temperature-sensitive point mutation in the *unc-52* gene. This strain is an established folding reporter reflecting an age-dependent decline in motility under permissive temperature (*Ben-Zvi et al., 2009*). We monitored *unc-52(ts)*-dependent paralysis on day 4 of adulthood in different genetic backgrounds. At this time-point, only 20% of the

*unc-52(ts)* animals were motile, whereas the *glp-1* mutation rescued *unc-52(ts)*-dependent paralysis in 80% of the animals (*Figure 3D,P*-value<0.001). Perturbation of the endo-siRNA pathway by a mutation in the *dcr-1* gene partially reduced the motility of *glp-1* mutants (*Figure 3D,P*-value<0.001). This suggests that endo-siRNAs are required for some aspects of correct protein folding in adult GSC(-) animals.

## Identification of endo siRNA-regulated genes in GSC(-) animals

We next set out to identify potential direct and indirect targets affected by endo-siRNA in GSC(-) animals. To this end, we compared both the mRNA transcriptomes and the siRNAs of GSC(-) animals in the presence or absence of the *dcr-1(mg375)* point mutation.

We predicted that direct mRNA targets of the endo-siRNA would be present at low levels in GSC(-) animals but stabilized when dicer activity is disrupted. We identified a group of 132 genes whose levels consistently and significantly increased by more than 1.5 fold (p-value<0.05) in *dcr-1 glp-1* double mutants compared to that of *glp-1* animals with wild-type dicer activity (*Supplementary file 4*). 84 of these genes were previously reported as regulated by endo-siRNAs (*Asikainen et al., 2007*; *Supplementary file 4*), attesting to the validity of our data. A protein-protein interaction network of these 132 genes using STRING (*Szklarczyk et al., 2019*) highlighted a group of 64 interacting genes whose interaction was based primarily on co-expression rather than on physical, genetic, or physiological interactions (*Figure 4—figure supplement 1*). Strikingly, 49 out of these 64 genes overlapped with the list of 84 *rrf-1*-regulated genes.

This group of 132 genes, whose transcripts are down-regulated (directly or indirectly) by endogenous siRNAs in *glp-1* mutants, are likely to include putative direct targets of the siRNA pathway in these animals. These direct targets are predicted to have increased levels of siRNA directed towards them in *glp-1* mutants compared to that of the dicer-defective double mutant. To identify these putative direct endo-siRNA targets, we generated small RNA libraries using a method that mainly captures secondary endo-siRNAs (*Fishman et al., 2018*) from *glp-1* and *dcr-1 glp-1* GSC(-) mutant animals, synchronized to day 1 of adulthood (*Figure 4A*). As expected, *glp-1* and *dcr-1 glp-1* mutants lacked 21u-RNAs, which are specifically expressed in the germ cells (*Wang and Reinke, 2008*; *Supplementary file 5*). We identified 138 genes whose endo-siRNA levels were ten-fold higher in *glp-1* compared to *dcr-1 glp-1* mutants (*Supplementary file 5*). This defines a group of endo-siRNAs whose production in GSC(-) animals relied on the integrity of the helicase domain of dicer. Among the list of genes targeted by these endo-siRNAs, the mRNA levels of five genes (ZK380.5, W04B5.1, ZK402.2, ZK402.3, F55C9.3) were decreased in a *dcr-1*-dependent manner in *glp-1* mutants. Thus, we considered these five genes as potential direct endo-siRNA targets in GSC(-) animals (*Figure 4A*, *Supplementary file 5*).

## Identification of endo-siRNA regulated genes that affect lifespan

Next, we set out to identify *dcr-1*-regulated genes whose silencing is required for the longevity of GSC(-) animals. Silencing of these genes in *dcr-1 glp-1* double mutants, by means other than endo-siRNA, should restore the extended lifespan and improve proteostasis to the typical levels as in GSC(-) animals. As the *dcr-1(mg375)* mutation does not compromise the exogenous RNAi pathway, we examined the effect of RNAi silencing of a select group of these genes on the lifespan of *dcr-1 glp-1* double mutants. Specifically, we focused on the five genes that may be directly targeted by endo-siRNA (identified by the overlap between their expression levels and the levels of their corresponding endo-siRNAs) (*Supplementary file 5*). In addition, we examined the requirement of 18 cellular protein modification-related genes, identified as significantly enriched within the group of 132 endo-siRNA repressed genes by DAVID enrichment analysis (p-value<0.01) (*Supplementary file 6*).

We treated *dcr-1 glp-1* double mutants with RNAi against each of these genes, and qualitatively examined the number of live animals in the plates on days 8–11. At these time-points, the survival of *dcr-1 glp-1* double mutants was significantly reduced compared to *glp-1* mutants. We found seven RNAi clones that improved the survival of *dcr-1 glp-1* mutants in the screen (*Supplementary file 7*). These included ZK402.2, ZK402.3 and W04B5.1, which might be direct endo-siRNA targets, and five modification-related genes (F26E4.5, F26A1.3, C24D10.1, C03C10.2, M05B5.1), which were probably indirectly regulated by endo-siRNAs. ZK402.2, and ZK402.3 are homologous 12.4 and 5.4 kDa proteins of unknown function. W04B5.1 is a pseudogene, whose expression is up-regulated in *rrf-3*

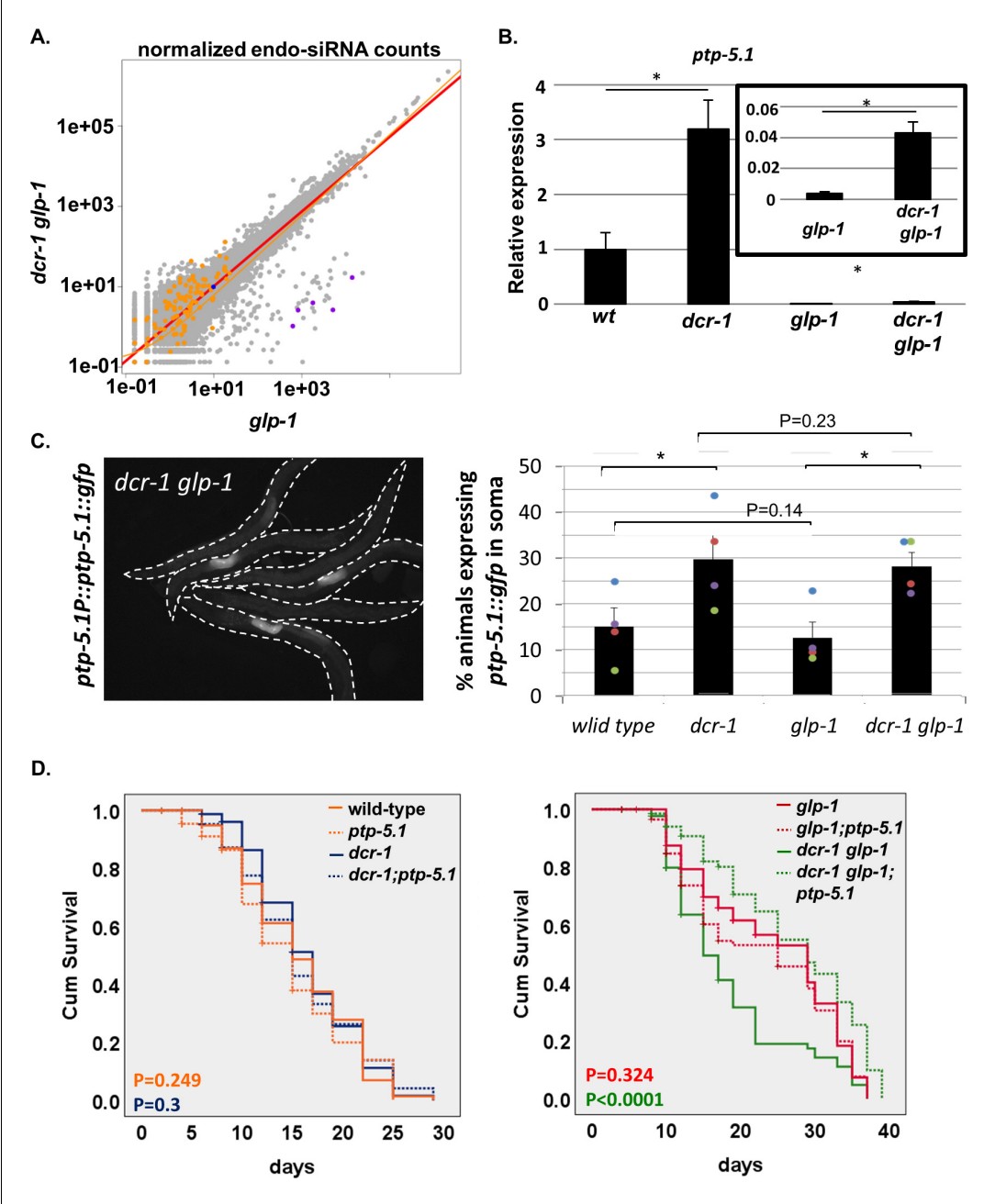

**Figure 4.** Inactivation of *ptp-5.1* restores longevity in GSC(-) animals with perturbed endogenous siRNA. (A) Scatter plot depicts comparisons of gene-by-gene siRNA counts from three paired *glp-1* mutant and *dcr-1 glp-1* double mutant samples. Gray- all genes. Orange- 132 genes whose levels increased by more than 1.5 folds in *dcr-1 glp-1* double mutants compared to *glp-1* single mutants at the transcriptome analysis. Purple- five overlapping genes between the transcriptome analysis and siRNA seq, which are candidate direct targets of endo-siRNA. Blue- *ptp-5.1* (*c24d10.1*). See also ***Figure 4—figure supplement 1*** and ***Supplementary files 4***, ***5***, ***6,*** and ***7***. (B) qRT-PCR of *ptp-5.1* transcript on day 1 of adulthood. Asterisks mark Student's t-test values of p value<0.05 of 4 independent experiments. Data are presented as mean ± SEM. The low levels of *ptp-5.1* transcript in GSC(-) animals is consistent with the interpretation that in GSC(+) animals, most of the transcript is expressed in the germline. (C) Representative fluorescent images of *dcr-1 glp-1* transgenic animals expressing an extrachromosomal array of a translational reporter fused to the *ptp-5.1* gene, driven by *ptp-5.1* upstream sequences. Note that the reporter is only detected in a fraction of the animals. Animals that expressed the reporter displayed a clear fluorescent signal specifically in two adjacent cells in the mid-intestine (see ***Figure 4—figure supplement 2B***). A similar expression pattern of the transgene was observed in all genetic backgrounds (see ***Figure 4—figure supplement 2***). Bars represent mean of the percentage of animals expressing the *ptp-5.1::gfp* transgene in the intestine. At least 250 animals per strain were scored in four independent experiments. Different colors represent independent experiments. Cochran-Mantel-Haenszel test P-values are indicated. Asterisks mark p<0.001. Somatic expression of the transgene was detected in all backgrounds in a fraction of the animals. Inactivation of the endo-siRNA pathway by the *dcr-1* mutation increased the

*Figure 4 continued on next page*

*Figure 4 continued*

fraction of the animals expressing the reporter in their mid-intestine. (D) *ptp-5.1(tm6122)* extended the lifespan of *dcr-1 glp-1* double mutants. Breslow (Generalized Wilcoxon) P-values for each mutant vs. the mutant; *ptp-5.1(tm6122)* double mutant are indicated. See *Supplementary file 1* for additional lifespan data.

The online version of this article includes the following figure supplement(s) for figure 4:

**Figure supplement 1.** String analysis of 132 genes whose levels increased by more than 1.5 fold in *dcr-1 glp-1* double mutants compared to *glp-1* mutant.

**Figure supplement 2.** The *ptp-5.1::gfp* transgene is expressed in the intestine of a fraction of the animals.

and *eri-1* mutants, in which endo-siRNA production is disrupted (*Gent et al., 2010*; *Pavelec et al., 2009*). The group of modification-related genes included a protein tyrosine phosphatase gene (*c24d10.1*), whose expression is up-regulated in *rrf-3(pk1426)* and *eri-1(mg366)* mutants (*Asikainen et al., 2007*). Given this established connection between the tyrosine phosphatase *c24d10.1* and the endo-siRNAs pathway, we further examined the role of its silencing in the longevity of *glp-1* mutants. For simplicity, we named *c24d10.1* as *ptp-5.1* (protein tyrosine phosphatase 5.1), based on its putative protein tyrosine phosphatase activity.

## *ptp-5.1* transcript levels are indirectly regulated by endo-siRNAs in GSC (-) animals

First, we used qRT-PCR to follow the transcript levels of *ptp-5.1* in the different genetic backgrounds. As previously reported (*Asikainen et al., 2007*), we found that *ptp-5.1* transcript levels increase upon interference with endo-siRNA processing in wild-type animals. In addition, *ptp-5.1* transcripts were almost absent in GSC(-) animals compared to wild-type controls (*Figure 4B*). This is consistent with previous work indicating that *ptp-5.1* may be a sperm-specific gene (*Ortiz et al., 2014*). Interestingly, we did detect a low level of the *ptp-5.1* transcript in germline-less *dcr-1 glp-1* double mutants (*Figure 4B* inset), implying that the *ptp-5.1* transcript is expressed to an extent in the soma of these animals (and perhaps also in the wild-type animals), and that the low levels of *ptp-5.1* transcript in the soma are regulated in an endo-siRNA dependent manner, at least in GSC(-) mutants.

To further analyze the somatic expression of *ptp-5.1*, we generated transgenic animals expressing an extra-chromosomal translational reporter of the PTP-5.1 protein fused to GFP, driven by the *ptp-5.1* promoter. Interestingly, whereas no detectable expression of the transgene was observed in the majority of the wild-type transgenic animals, we did detect clear expression of the transgene in 15% of the animals (*Figure 4C*, *Figure 4—figure supplement 2*). In most of these animals, the reporter was expressed in two adjacent cells in the middle of the intestine (*Figure 4—figure supplement 2*). In very few cases, we detected expression of the transgene in the animals' nerve system, in lieu of the intestine. Interference with endo-siRNA processing, by the *mg375* mutation in the *dcr-1* helicase domain, increased the fraction of animals with the intestinal expression of the reporter. Interestingly, removal of the germline did not further alter the somatic expression of the reporter (i.e. 15% of the germline-less animals expressed the transgene in two of their intestinal cells, and twice as many germline-less animals expressed the transgene upon interference with the endo-siRNA pathway, *Figure 4C*). Altogether, these findings support the notion that *ptp-5.1* expression in the soma is suppressed in an endo siRNA-dependent manner in wild-type animals and in GSC(-) animals. We assume that this expression corresponds to the product of a very small fraction of the *ptp-5.1* transcripts, the majority of which are expressed in the germline rather than in the soma (*Figure 4B*). Nevertheless, since we did not detect a change in the amount of *ptp-5.1* endo-siRNAs between the *glp-1* and the *dcr-1 glp-1* samples (fold change = 1, P-adj = 0.278, *Figure 4A*), we conclude that the regulation of *ptp-5.1* transcript levels by endo-siRNAs in *glp-1* mutants is indirect.

## Inactivation of *ptp-5.1* restores longevity and improves proteostasis in GSC(-) animals with perturbed endo-siRNA

Our limited RNAi screen, described above, suggested that more *dcr-1 glp-1* double mutants were alive upon treatment with *ptp-5.1* RNAi. Hence, we examined how a deletion mutation in the *ptp-5.1* gene affected the lifespan of *dcr-1 glp-1* mutants, in a detailed lifespan experiment. We found

that a mutation in *ptp-5.1* extended the lifespan of *dcr-1 glp-1* double mutants to the level of that of long-lived GSC(-) *glp-1* mutants (*Figure 4D*). In contrast, the *ptp-5.1* mutation did not significantly change the lifespan of wild-type animals, *dcr-1* single mutants or *glp-1* single mutants (*Figure 4D* and *Supplementary file 1*). Thus, *ptp-5.1* seems to specifically limit the lifespan of GSC(-) animals when the endo-siRNA pathway is compromised.

Since we found that endo-siRNAs were required for HSF-1's activity in GSC(-) animals, we also examined the effect of *ptp-5.1* inactivation on the compromised heat shock response of GSC(-) *dcr-1 glp-1* double mutants. We found that mutation in *ptp-5.1* fully restored the heat shock resistance of *dcr-1 glp-1* double mutants to the same survival level as that of *glp-1* single mutants. In contrast, introduction of a mutation in *ptp-5.1* into wild-type or *dcr-1* animals did not alter their survival after heat shock (*Figure 5A*).

In addition to their sensitivity to heat shock, *dcr-1 glp-1* double mutants were less capable of forming HSF-1 foci within the nuclei in response to heat shock compared to *glp-1* animals (*Figure 2C*). Hence, we examined the ability of HSF-1 to form intra-nuclear foci upon heat shock in *dcr-1 glp-1* animals in the presence of the *ptp-5.1* mutation. We observed a significant increase in the average number of HSF-1-labeled foci per hypodermal cell in heat-shocked *dcr-1 glp-1; ptp-5.1* triple mutants compared to *dcr-1 glp-1* double mutants. This increase in the number of foci upon *ptp-5.1* inactivation was unique to *dcr-1 glp-1* mutants, as its inactivation did not increase the

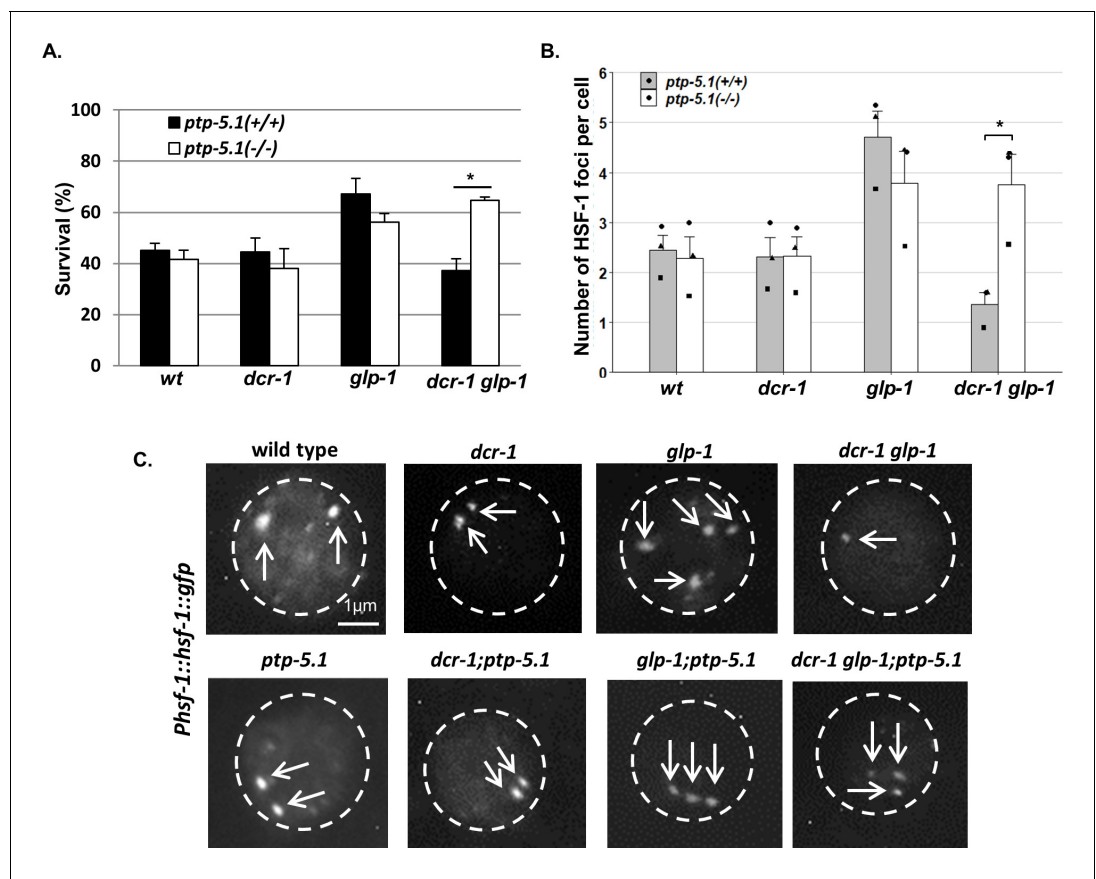

**Figure 5.** Inactivation of *ptp-5.1* improves proteostasis in GSC(-) animals with perturbed endogenous siRNA. (**A**) Thermo-resistance of age-synchronized animals subjected to heat shock (37°C, 9 hr) on day 2 of adulthood upon 5 hr of recovery at 25°C (120 animals per treatment, *N* = 3). Asterisks mark Cochran-Mantel-Haenszel test values of p<0.001. (**B**) Bars represent mean of mean number of HSF-1::GFP nuclear foci per hypodermal cell. At least 140 cells per genotype were scored in a total of 3 independent experiments. Dots represent mean number of HSF-1 foci per cell with different shapes representing independent experiments. Asterisk marks p-value<0.05 determined by One-Way ANOVA followed by Tukey's post hoc analysis. Data are presented as mean ± SEM. See *Supplementary file 8* for statistic details. (**C**) Fluorescence micrographs of representative hypodermal cells in day three adults, harboring a single copy of the *Phsf-1::hsf-1::gfp* transgene upon exposure to heat shock. Nuclear boundaries are circled. Exposures and contrast were adjusted for each picture independently to best emphasize foci amount.

number of foci in wild-type, *dcr-1* mutants or *glp-1* mutants (*Figure 5B–C*). Altogether, these experiments suggest that *ptp-5.1* downregulation is sufficient for restoration of the longevity and the improved heat shock response, which were compromised by the endo-siRNA-deficiency in GSC(-) animals.

## Discussion

Although the idea that the rate of aging can be slowed down and the existence of longevity-promoting genes and pathways are well established, we are still deciphering the underlying mechanism whereby longevity pathways extend lifespan. At the molecular level, most life-span extension pathways involve extensive remodeling of the transcriptome. Accordingly, genes involved in chromatin modifications (*Maures et al., 2011*; *Greer et al., 2011*), RNA modifying pathways (*Heintz et al., 2017*; *Tabrez et al., 2017*; *Son et al., 2017*; *Masse et al., 2008*), a long list of transcription factors, as well as several miRNAs are all required for the longevity of animals (*Denzel et al., 2019*).

Besides microRNAs, *C. elegans* produces additional small RNAs as well as long noncoding RNAs, targeting coding genes, pseudogenes, and transposons. These, too, could potentially alter gene expression landscape and affect basic biological processes such as lifespan. Although advances in sequencing technologies have led to the identification of thousands of endo-siRNAs, their biological impact is not fully understood. In this study, we demonstrate that endo-siRNAs are implicated in longevity regulation in animals reprogrammed to slow down aging due to the depletion of their germline.

The processing of endo-siRNAs and the silencing of their target genes are complex. Endo-siRNA processing in *C. elegans* requires multiple proteins, including dicer, several RdRPs, and different Argonaute proteins (*Yigit et al., 2006*; *Duchaine et al., 2006*; *Fischer et al., 2011*). We found that impairment of primary siRNA production (via *dcr-1* and *rrf-3* mutations), impairment of secondary siRNA production and target silencing in the cytoplasm (via *ergo-1* mutation), and impairment of target silencing in the nucleus (via *nrde-3* mutation), all compromise longevity associated with germline depletion. Interestingly, impairment of the endo-siRNA machinery did not consistently affect the lifespan of wild-type animals (*Figure 1A–E*, *Supplementary file 1*). This suggests that the silencing by endo-siRNAs affects target genes that specifically limit longevity, at least in GSC(-) animals, rather than normal lifespan. Furthermore, given the lack of germline in these animals, this establishes an important somatic role for endo siRNAs, which have been mostly associated with germline inheritance (*Rechavi and Lev, 2017*; *Rechavi et al., 2014*; *Kishimoto et al., 2017*; *Ni et al., 2016*).

Depending on the mutated endo siRNA-related gene, different extents of lifespan shortening were observed in *glp-1* mutants. The strongest effect on lifespan was observed with the *nrde-3* mutant. This suggests that the silencing event that controls the longevity of GSC(-) animals is mediated via nuclear silencing. We attribute the weaker effects of the other endo-siRNA related mutations to partial inactivation of the endo-siRNA processing, due to possible redundancies (for example in the case of mutation in the *rrf-3* RdRP, which is one of four RdRPs in *C. elegans*), or due to mis-regulation of only a subset of endo siRNAs (as in the case of the point mutation in the helicase domain of dicer (*Welker et al., 2010*).

To identify longevity-related endo siRNA regulated genes and pathways in GSC(-) animals, we undertook a functional genomic approach, monitoring both the abundance of all endo siRNAs as well as the transcriptomic changes in the same animals. This genomic approach was then complemented by functional studies leading to the identification of direct and indirect endo siRNA-regulated genes, whose endo siRNA-dependent silencing is critical for the longevity of GSC(-) animals.

Among the identified genes were putative direct endo siRNA targets genes as well as the indirect target *ptp-5.1*. In addition, the transcriptome analysis pointed out a set of HSF-1 regulated genes, primarily but not exclusively chaperone genes, whose levels were reduced in *dcr-1 glp-1* mutants, suggesting that the endo siRNA protects the cytosolic heat shock response in *glp-1* mutants (*Figure 2—figure supplement 1*, *Figure 4—figure supplement 1*). The impairment in HSF-1 activity in endo siRNA defective *glp-1* mutants could in turn account for the compromised longevity and proteostasis in these animals, as both the longevity and the superior proteostasis of *glp-1* mutants are dependent on HSF-1 (*Shemesh et al., 2013*; *Hansen et al., 2005*). These findings suggest that endo siRNAs are required to remove an activity restraint on the proteostasis-related transcription factor HSF-1 in *glp-1* mutants. Interestingly, the requirement to counteract restraining pathways to

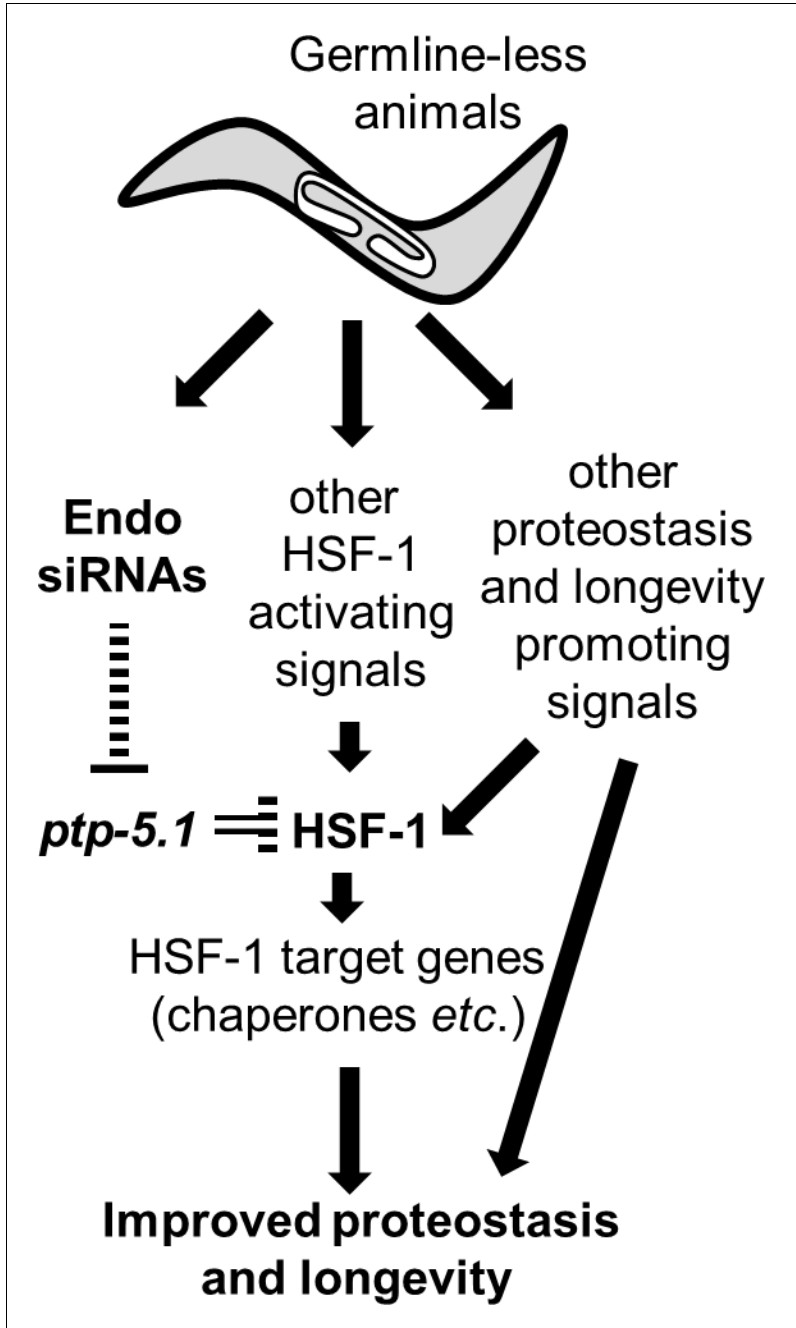

**Figure 6.** Endo-siRNAs improve proteostasis and promote longevity of GSC(-) animals by enabling HSF-1 activation. Model: Germline-less animals extensively remodel their transcriptome to promote longevity and proteostasis. HSF-1 is one of the central transcription factors that transcribe proteostasis and longevity-promoting genes. We find that endo-siRNAs are critical for HSF-1 activity in GSC(-) animals, and consequently for their longevity and improved proteostasis. These endo-siRNAs are important because they indirectly limit the level of the tyrosine phosphatase *ptp-5.1*. The inhibition of this tyrosine phosphatase is critical for HSF-1 activation in proteostasis challenging settings such as heat-shock and aging. Whereas the release of HSF-1 from *ptp-5.1* inhibition is required for the proteostasis and longevity benefits in GSC(-) animals, it is not sufficient. To achieve effective remodeling of the proteostasis and longevity promoting networks, germline removal must coordinate between the removal of *ptp-5.1*-dependent inhibition of HSF-1 and additional cellular events that promote HSF-1 activity such as reducing the repressive chromatin marks at HSF1-regulated stress-responsive genes.

promote HSF-1 activity is also observed in animals with reduced insulin/IGF-1 signaling, in which HSF-1 must dissociate from inhibitory factors, such as DDL-1, to gain activation (*Chiang et al., 2012*). These mechanisms are consistent with the multi-step nature of HSF-1 activation (*Sarge et al., 1993*).

Of the different endo-siRNA regulated genes identified in this study, we focused on tyrosine phosphatase *ptp-5.1*. We found that endo siRNA indirect repression of *ptp-5.1* in the soma is critical for HSF-1 activation in GSC(-) animals. This places *ptp-5.1* as an upstream inhibitor of HSF-1 in GSC(-) animals. Interestingly, expression data analysis indicates that *ptp-5.1* is primarily expressed in the sperm and is normally absent from the soma (*Ortiz et al., 2014*). Our data confirm that the majority of the *ptp-5.1* transcripts reflect germline expression rather than somatic expression. Nevertheless, we did detect PTP-5.1 expression in the soma in a subset of the transgenic animals (*Figure 4C*). Hence, the removal of this low amount of *ptp-5.1* in the soma is critical for the proteostasis maintenance and the longevity of GSC(-) animals.

Inactivation of *ptp-5.1* completely restored both longevity and the superior proteostasis of *dcr-1 glp-1* mutants to the level of *glp-1* mutants, implying that it is a proteostasis and longevity-limiting factor (*Figure 4D* and *Figure 5*). Nevertheless, *ptp-5.1* mutants, which do not express *ptp-5.1* in the soma, are not long-lived or heat shock resistant as are GSC(-) animals (*Figure 4D* and *Figure 5*). This suggests that in addition to the removal of the proteostasis/lifespan limiting gene *ptp-5.1* from the soma, additional proteostasis/lifespan promoting events must take place to obtain these benefits. For example, in GSC(-) animals, *jmjd-3.1* relaxes the chromatin and allows more efficient binding of HSF-1 to the DNA in mature GSC(-) animals (*Labbadia and Morimoto, 2015*). Thus, a combination of HSF-1 activating pathways along with the removal of HSF-1 inhibitory pathways is concomitantly required to activate HSF-1 to promote proteostasis and longevity (see model in *Figure 6*).

In summary, this study provides insight into the molecular mechanisms that enable enhanced proteostasis and longevity in GSC(-) animals. Specifically, this study identified a set of endo siRNA-regulated longevity-limiting genes, whose expression must be repressed in the context of the longevity treatment to enable the activation of the heat shock transcription factor HSF-1. This establishes a role for endo siRNAs in the regulation of proteostasis and aging in long-lived germline-less animals. We propose that such longevity-limiting genes and pathways may provide new targets for interventions that, along with pro-longevity treatments, may more effectively attenuate or reverse systemic dysfunction associated with age, and therefore have the potential to reduce overall disease risk for chronic and proteostasis-related diseases in the elderly.

## Materials and correspondence

Correspondence and material requests should be addressed to Sivan Henis-Korenblit.

## Materials and methods

**Key resources table**

| Reagent type (species) or resource | Designation | Source or reference | Identifiers | Additional information |
|---|---|---|---|---|
| Strain, strain background (*C. elegans*) | N2 | Caenorhabditis Genetics Center | Wild Type | |
| Strain, strain background (*C. elegans*) | CF1903 | Caenorhabditis Genetics Center | *glp-1(e2144)* | outcrossed three times in C Kenyon's lab |
| Strain, strain background (*C. elegans*) | YY470 | Caenorhabditis Genetics Center | *dcr-1(mg375)* | an outcrossed version of YY11 *dcr-1(m9375)* |
| Strain, strain background (*C. elegans*) | SHK77 | This paper | *dcr-1(mg375) glp-1(e2144)* | Strain created in S Henis-Korenblit lab |
| Strain, strain background (*C. elegans*) | CF3152 | Cynthia Kenyon lab | *rrf-3(pk1426)* | outcrossed three times in C Kenyon's lab |

*Continued on next page*

*Continued*

| Reagent type (species) or resource | Designation | Source or reference | Identifiers | Additional information |
|---|---|---|---|---|
| Strain, strain background (C. elegans) | SHK55 | This paper | rrf-3(pk1426); glp-1(e2144) | Strain created in S Henis-Korenblit lab |
| Strain, strain background (C. elegans) | SHK80 | This paper | ergo-1(gg98) | Strain outcrossed two times in S Henis-Korenblit lab. Total eight outcrosses |
| Strain, strain background (C. elegans) | SHK87 | This paper | glp-1(e2144); ergo-1(gg98) | Strain created in S Henis-Korenblit lab |
| Strain, strain background (C. elegans) | YY158 | Caenorhabditis Genetics Center | nrde-3(gg66) | |
| Strain, strain background (C. elegans) | SHK328 | This paper | glp-1(e2144); nrde-3(gg66) | Strain created in S Henis-Korenblit lab |
| Strain, strain background (C. elegans) | SHK53 | This paper | sid-1(pk3321) | Strain outcrossed four times in S Henis-Korenblit lab |
| Strain, strain background (C. elegans) | SHK56 | This paper | glp-1(e2144); sid-1(pk3321) | Strain created in S Henis-Korenblit lab |
| Strain, strain background (C. elegans) | OG497 | Caenorhabditis Genetics Center | unc-119(ed3);drSi13 [hsf-1p::hsf-1::GFP:: unc-54utr;Cb-unc-119+] | |
| Strain, strain background (C. elegans) | SHK299 | This paper | unc-119(ed3);drSi13 [hsf-1p::hsf-1::GFP:: unc-54utr;Cb-unc-119+]; glp-1(e2144) | Strain created in S Henis-Korenblit lab |
| Strain, strain background (C. elegans) | SHK300 | This paper | unc-119(ed3);drSi13 [hsf-1p::hsf-1::GFP:: unc-54utr;Cb-unc-119+]; dcr-1(mg375) | Strain created in S Henis-Korenblit lab |
| Strain, strain background (C. elegans) | SHK301 | This paper | unc-119(ed3);drSi13 [hsf-1p::hsf-1::GFP:: unc-54utr;Cb-unc-119+]; dcr-1(mg375) glp-1(e2144) | Strain created in S Henis-Korenblit lab |
| Strain, strain background (C. elegans) | AM140 | Caenorhabditis Genetics Center | rmIs132 [unc-54p::Q35::YFP] | |
| Strain, strain background (C. elegans) | SHK409 | This paper | rmIs132 [unc-54p::Q35::YFP]; glp-1(e2144) | Strain created in S Henis-Korenblit lab |
| Strain, strain background (C. elegans) | SHK412 | This paper | rmIs132 [unc-54p::Q35::YFP]; dcr-1 (mg375) | Strain created in S Henis-Korenblit lab |
| Strain, strain background (C. elegans) | SHK410 | This paper | rmIs132 [unc-54p::Q35::YFP]; dcr-1 (mg375) glp-1(e2144) | Strain created in S Henis-Korenblit lab |
| Strain, strain background (C. elegans) | HE250 | Caenorhabditis Genetics Center | unc-52(e669su250) | |
| Strain, strain background (C. elegans) | SHK574 | This paper | unc-52(e669su250); glp-1(e2144) | Strain created in S Henis-Korenblit lab |
| Strain, strain background (C. elegans) | SHK575 | This paper | unc-52(e669su250); dcr-1(mg375) | Strain created in S Henis-Korenblit lab |

*Continued on next page*

*Continued*

| Reagent type (species) or resource | Designation | Source or reference | Identifiers | Additional information |
|---|---|---|---|---|
| Strain, strain background (*C. elegans*) | SHK576 | This paper | *unc-52(e669su250); dcr-1(mg375) glp-1(e2144)* | Strain created in S Henis-Korenblit lab |
| Strain, strain background (*C. elegans*) | SHK415 | This paper | *ptp-5.1(tm6122)* | Strain outcrossed three times in S Henis-Korenblit lab |
| Strain, strain background (*C. elegans*) | SHK470 | This paper | *glp-1(e2144); ptp-5.1(tm6122)* | Strain created in S Henis-Korenblit lab |
| Strain, strain background (*C. elegans*) | SHK469 | This paper | *dcr-1(mg375); ptp-5.1(tm6122)* | Strain created in S Henis-Korenblit lab |
| Strain, strain background (*C. elegans*) | SHK471 | This paper | *dcr-1(mg375) glp-1(e2144); ptp-5.1(tm6122)* | Strain created in S Henis-Korenblit lab |
| Strain, strain background (*C. elegans*) | SHK405 | This paper | *unc-119(ed3);drSi13[hsf-1p::hsf-1::GFP::unc-54utr; Cb-unc-119+];glp-1(e2144); ptp-5.1(tm6122)* | Strain created in S Henis-Korenblit lab |
| Strain, strain background (*C. elegans*) | SHK406 | This paper | *unc-119(ed3);drSi13[hsf-1p::hsf-1::GFP::unc-54utr; Cb-unc-119+];dcr-1(mg375); ptp-5.1(tm6122)* | Strain created in S Henis-Korenblit lab |
| Strain, strain background (*C. elegans*) | SHK407 | This paper | *unc-119(ed3);drSi13[hsf-1p::hsf-1::GFP::unc-54utr; Cb-unc-119+];dcr-1(mg375) glp-1(e2144);ptp-5.1(tm6122)* | Strain created in S Henis-Korenblit lab |
| Strain, strain background (*C. elegans*) | SHK619 | This paper | *biuEx63[Pptp-5.1::genomic ptp-5.1::gfp+rol-6]* | Strain created in S Henis-Korenblit lab |
| Strain, strain background (*C. elegans*) | SHK622 | This paper | *dcr-1(mg375); biuEx63[Pptp-5.1::genomic ptp-5.1::gfp+rol-6]* | Strain created in S Henis-Korenblit lab |
| Strain, strain background (*C. elegans*) | SHK623 | This paper | *glp-1(e2144); biuEx63[Pptp-5.1::genomic ptp-5.1::gfp+rol-6]* | Strain created in S Henis-Korenblit lab |
| Strain, strain background (*C. elegans*) | SHK624 | This paper | *glp-1(e2144); biuEx63[Pptp-5.1::genomic ptp-5.1::gfp+rol-6]* | Strain created in S Henis-Korenblit lab |
| Strain, strain background (*C. elegans*) | SHK620 | This paper | *dcr-1(mg375) glp-1(e2144); biuEx63[Pptp-5.1::genomic ptp-5.1::gfp+rol-6]* | Strain created in S Henis-Korenblit lab |
| Strain, strain background (*C. elegans*) | SHK621 | This paper | *dcr-1(mg375) glp-1(e2144); biuEx63[Pptp-5.1::genomic ptp-5.1::gfp+rol-6]* | Strain created in S Henis-Korenblit lab |
| Sequence-based reagent | *act-1* FW | This paper | qPCR primers | CCAATCCAAGAGA GGTATCCTTAC |
| Sequence-based reagent | *act-1* BW | This paper | qPCR primers | CATTGTAGAAGGT GTGATGCCAG |
| Sequence-based reagent | *F44E5.5* FW | This paper | qPCR primers | CAGAATGGAAAGGT TGAGATCCTCGCC |
| Sequence-based reagent | *F44E5.5* BW | This paper | qPCR primers | ACTGTATTCTCTGGAT TACGAGCTGCTTGA |
| Sequence-based reagent | *hsp-16.2* BW | This paper | qPCR primers | CTCTCCATCTGAGTCT TCTGAGATTGTTAACA |
| Sequence-based reagent | *hsp-16.2* FW | This paper | qPCR primers | CAATTCTTGTTCTC CTTGGATTGATAGCGT |

*Continued on next page*

*Continued*

| Reagent type (species) or resource | Designation | Source or reference | Identifiers | Additional information |
|---|---|---|---|---|
| Sequence-based reagent | *hsp-12.6 BW* | This paper | qPCR primers | GATGGAGTTGTCA ATGTCCTCGACGAC |
| Sequence-based reagent | *hsp-12.6 FW* | This paper | qPCR primers | TTGTGCTCCATATGGA TTTCAAGAAGTTCTCC |
| Sequence-based reagent | *ptp-5.1 FW* | This paper | qPCR primers | AAGGCTCCGTC TCCTGCACT |
| Sequence-based reagent | *ptp-5.1 BW* | This paper | qPCR primers | TCCAGAGACACTTG TTGCTATCGGAG |
| Sequence-based reagent | *bw_kpni_ptp-5.1_cds* | This paper | cloning primers | GACAATGGTACCTTTCC AGGTCCCATCATACT |
| Sequence-based reagent | *fw_PstI_ptp-5.1_Prom* | This paper | cloning primers | ATGCCTGCAGCACC TACATTACGCCTGCGC |
| Antibody | anti-HSF-1, rabbit polyclonal Antibody | Abcam | ABE1044 | WB(1:1,000) |
| Antibody | anti-Tubulin mouse monoclonal ascites fluid B-5-1-2 | SIGMA-ALDRICH | T5168 | WB(1:6000) |
| Antibody | anti-Tubulin, mouse monoclonal | DHSB | AA4.3 | WB(1:2,000), RRID:AB_579793 |
| Commercial kit | RNA spike-in kit | Agilent | 5188–5279 | |
| Commercial kit | miRVana miRNA isolation kit (w/phenol) | Ambion | AM1560 | |
| Commercial assay | *C. elegans* microarray 4 × 23,000 | Agilent | G2519F-020186 | |
| Chemical compound | TRIzol | Ambion | 15596026 | |
| Chemical compound | Linoleic acid sodium salt | Sigma | L8134 | |
| Chemical compound | Maxima SYBR GREEN | Thermo Scientific | K0221 | |
| Instrument | microarray scanner | Agilent | G2565BA | |
| Instrument | CFX-96 real time system | BioRad | | |
| Software, algorithm | Agilent Feature Extraction software | Agilent | version 9.5.1.1 | Agilent Technologies, RRID:SCR_014963 |
| Software, algorithm | Partek Genomics Suite software | Partek | version 6.6 | RRID:SCR_011860 |
| Software, algorithm | DAVID | | | RRID:SCR_001881 |
| Software, algorithm | STRING | | | RRID:SCR_005223 |
| Software, algorithm | SPSS | SPSS | | RRID:SCR_002865 |

## Molecular cloning and generation of transgenic animals

The genomic fragment of *ptp-5.1*, including the coding region and 2 kb upstream sequence, was amplified from the corresponding cosmid and cloned into the *PstII* and *KpnI* sites in the L3691 plasmid, in frame with the GFP coding sequence. Germline transformations were performed by injection of 40 ng/µl plasmid and 60 ng/µl of *rol-6(su1006)* as a co-transformation marker into wild-type animals. Transgenic animals were allowed to lay eggs for 4 hrs. Eggs were raised at 25 degrees until day 1 of adulthood. On day 1 of adulthood, the animals were anesthetized on 2% agarose pads containing 2 mM levamisol. Number of GFP-expressing animals was scored with 100X magnification. At least 250 animals per strain were scored in four independent experiments. For localization analysis, DAPI staining was performed using an acetone-based protocol, which preserves the GFP signal, as

previously described (*Liang et al., 2018*). Images were taken with a CCD digital camera using a Nikon 90i fluorescence microscope and merged using ImageJ.

## RNA interference

Bacteria expressing dsRNA were cultured overnight in LB containing 10 µg/mL tetracycline and 100 µg/mL ampicillin. Bacteria were seeded on NGM plates containing 2 mM IPTG and 0.05 mg/ml carbenicillin. RNAi clone identity was verified by sequencing. Eggs were placed on plates and synchronized at day 0 (L4).

## Lifespan and paralysis assay

Eggs were placed on standard NGM media with OP50 bacteria. Lifespan was scored every 1–2 days. Animals were raised at 25 degrees from eggs until day 1, and transferred to 20 degrees henceforth, except for the DGLA related lifespans. Related lifespans experiments were performed concurrently to minimize variability. In all experiments, lifespan was scored as of the L4 stage, which was set as t = 0. Animals that ruptured or crawled off the plates were included in the lifespan analysis as censored worms. For DGLA-supplemented lifespans, linoleic acid sodium salt (Sigma L8134) was dissolved in water and added to 0.1% NP-40-containing plates to a final concentration of 150 µM. Plates containing the detergent NP-40 (0.1%) were used as control. DGLA related lifespans were performed at 20 degrees from egg stage. SPSS program was used to determine the means and the P-values. P-values were calculated using the Breslow (Generalized Wilcoxon) method (*Gehan, 1965*).

## Microarray analysis

Total RNA was extracted with TRIzol reagent (Ambion, 15596026) from wild type, *dcr-1(mg375), glp-1(e2144), and dcr-1(mg375) glp-1(e2144)* animals. RNA concentrations were measured using a NanoDrop spectrophotometer (ND-1000), and sample quality was checked using a bioanalyzer (Agilent). 200 ng of total RNA of each sample, in the presence of control RNAs (RNA spike-in kit, Agilent), was labeled with either Cy-3 or Cy-5 using the low-input quick amp labeling kit, two-color (Agilent) following the manufacturer's protocol. Each strain had four biological replicates. Equal amounts of labeled RNA were hybridized overnight to Agilent's *C. elegans* microarray 4 × 23,000 at 60°C. Hybridization mixes were prepared using the gene expression hybridization kit of Agilent following the manufacturer's protocol. Following hybridization, the each slide was first washed with Gene Expression Wash Buffer 1 (Agilent) and then with Gene Expression Wash Buffer 2 (Agilent). This was followed by an acetonitrile wash. Finally, the slides were placed in stabilization and drying solution (Agilent). The washed slides were scanned on an Agilent G2565BA microarray scanner. The data of all the arrays were first subjected to background correction and LOESS within-array normalization using Agilent Feature Extraction software (version 9.5.1.1, Agilent Technologies). The remaining analyses were performed in Partek Genomics Suite software (version 6.6, Partek, Inc). The log expression ratios were produced during the normalization step. Data from the four biological replicates were used to perform two-way ANOVA analysis. Genes with significantly up- or downregulated expression (p-value 0.05) were identified, with a cut-off of at least a 1.5-fold change. We then focused on expressed genes or pathways enriched as indicated by DAVID enrichment analysis program [22] and the gene ontology (GO) classification analysis. The DAVID GO fold change is defined as the ratio between the proportions of the submitted list and the proportion of the background one. Raw and processed data were deposited under the Gene Expression Omnibus, with accession number GSE122457.

## Quantitative RT-PCR

Animals were raised at 25 degrees until day 1. On day 1 of adulthood, total RNA was extracted with TRIzol reagent (Ambion, 15596026). RNA extraction, purification, and reverse transcription were carried out using standard protocol. Real-time PCR was done using Maxima SYBR GREEN (Thermo Scientific, K0221) in Step one plus instrument. Purified DNA templates were amplified in a BioRad CFX-96 real-time system. mRNA levels of *act-1* were used for normalization. P-values were calculated using Student's T-test.

## Small RNA sequencing and analysis

Animals were raised at 25 degrees until day 1 of adulthood. Low molecular RNA fraction was extracted using miRVana miRNA isolation kit (Ambion). Library preparation was done using QsRNA-seq protocol (*Fishman et al., 2018*). To allow ligation of secondary siRNA to adapter, two out of three phosphates were enzymatically removed from the 5'-termini, resulting in secondary siRNA enriched libraries. Sequences were processed as previously described (*Fishman et al., 2018*). This included demuxing, trimming, and collapsing the sequences. Only sequences longer than 14nt long were processed. The sequences were aligned to WS220 annotated genes (www.wormbase.org) using Bowite aligner (with parameters -v 0 -e 120 -a –strata –best) (*Langmead et al., 2009*). DEseq (*Anders and Huber, 2010*) package in R (http://www.r-project.org, with pooled-CR parameter) was used to evaluate siRNA expression. Genes with significant changes in DEseq normalized siRNA counts between *glp-1* mutant and *dcr-1 glp-1* mutant were considered if there was at least 10 fold difference with adjusted p-value after Benjamini–Hochberg correction (P-adj) <0.05) (*Supplementary file 5*). Raw and processed data were deposited under the Gene Expression Omnibus, with accession number GSE128935.

## Western blot

100 animals were boiled in protein sample buffer containing 2% SDS. Proteins were separated using standard PAGE separation, transferred to a nitrocellulose membrane, and detected by western-blotting using the following antibodies: anti-HSF-1 (ABE1004), anti-Tubulin (DHSB, 1:5000).

## HSF-1::GFP foci

HSF-1::GFP foci were scored as previously described (*Morton and Lamitina, 2013*). Specifically, eggs of animals containing HSF-1::GFP were placed on standard NGM media with OP50 bacteria and raised at 25 degrees until day 3 of adulthood. On day 3 of adulthood, the animals were moved to pre-warmed plates and exposed to heat shock (37°C for 10 min). Animals were anesthetized on 2% agarose pads containing 2 mM levamisol. Number of foci were scored immediately in posterior hypodermal cells (per experiment - seven nuclei assessed per worm, seven worms per strain) with 630X magnification. Foci scoring was done in the plane that showed the maximal number of foci per hypodermal cell. Images were taken with a CCD digital camera using a Nikon 90i fluorescence microscope. For each trial, exposure time was calibrated to minimize the number of saturated pixels and kept constant through the experiment. Exposures and contrast were adjusted for each picture independently to best emphasize foci amount. At least 140 cells per genotype were scored in a total of 3 independent experiments.

## Thermo-resistance assay

Age-synchronized animals (n > 40) were grown at 25°C until day 2 of adulthood. On day 2 of adulthood, animals were subjected to 37°C heat shock for 9 hr and recovered at 25°C for 5 hr. Animals that failed to move in response to a gentle touch with a metal pick were scored as dead.

## Motility assay

Age-synchronized animals (n > 30) that express *Punc-54::Q35::YFP (Q35)* were grown at 25°C until day 5 of adulthood. On day 5 of adulthood, animals were placed in 96 wells containing M9 buffer. Each animal was monitored visually over 15 s for trashing. Values are presented as bends per minute.

## Stiff body paralysis assay

Age-synchronized (n > 30) *unc-52(ts)* mutant animals were grown at 25°C until day 1 of adulthood. Animals were then shifted to 15°C, and paralysis was scored on day 4 of adulthood.

## RNAi screen

*dcr-1 glp-1* eggs were treated with RNAi against the listed genes until day 1. Viability of the animals in the plates was qualitatively scored at days 9–11 as improved or not improved. RNAi identity was verified by sequencing. Genes whose RNAi identity was not supported by sequencing were noted as no RNAi. Genes whose RNAi interfered with animal development were indicated as lethal.

## Statistical analysis

Error bars represent the standard error of the mean (SEM), unless noted otherwise. For qRT-PCR, P-values were calculated using the unpaired Student's t test. For lifespan experiments, P-values were calculated using the Breslow (Generalized Wilcoxon) method. For thermo-resistance assay and *unc-52(ts)* paralysis, P-values were calculated using the Cochran-Mantel-Haenszel test. For Q35 motility assay and HSF-1 foci formation assay, P-values were calculated using one-way ANOVA followed by Tukey's post hoc analysis. For Western Blot analysis, One sample Test and One-way ANOVA followed by Tukey's post hoc analysis were applied. For fraction of animals expressing the *ptp-5.1* transgene, P-values were calculated using the Cochran-Mantel-Haenszel test.

## Acknowledgements

We thank Dr. Jennifer Israel Cohen Benichou for statistical analysis and data presentation and Ms. Yael Laure for English editing. We thank members of the Henis-Korenblit laboratory for helpful discussions. We thank Dr. Shohei Mitani (National Bioresource Project for the nematode, Tokyo Women's Medical University School of Medicine, Japan) and the Caenorhabditis Genetics Center for providing nematode strains. This work was supported by funds from ISF grants no. 1571/15 and 689/19 to S.H.K. and no. 927/18 to ATL, by grant no. 3–12066 from the Israeli Ministry of Science, Technology and Space to S.H.K, and by the Israeli Centers of Research Excellence (I-CORE) program (Center No. 1796/12 to ATL).

## Additional information

### Funding

| Funder | Grant reference number | Author |
|---|---|---|
| Israel Science Foundation | 689/19 | Sivan Henis-Korenblit |
| Israel Science Foundation | 927/18 | Ayelet T Lamm |
| Israel Ministry of Science, Technology and Sports | 3-12066 | Sivan Henis-Korenblit |
| Israeli Centers for Research Excellence | 1796/12 | Ayelet T Lamm |

The funders had no role in study design, data collection and interpretation, or the decision to submit the work for publication.

### Author contributions

Moran Cohen-Berkman, Conceptualization, Data curation, Formal analysis, Investigation, Methodology, Writing - original draft; Reut Dudkevich, Conducted some of the lifespan experiments and some of the western blot experiments; Shani Ben-Hamo, Conducted some of the lifespan experiments and generated the dcr-1 glp-1 double mutant; Alla Fishman, Supervision, Methodology, Writing - review and editing, Contributed to the acquisition of the small RNA data; Yehuda Salzberg, Generated the transgenic animals; Hiba Waldman Ben-Asher, Formal analysis, H.W.B-A analyzed and interpreted the genomic data; Ayelet T Lamm, Formal analysis, Supervision, Investigation, Methodology, Writing - review and editing, A.T.L. analyzed and interpreted the small RNA data; Sivan Henis-Korenblit, Conceptualization, Formal analysis, Supervision, Funding acquisition, Investigation, Methodology, Writing - original draft, Project administration, Writing - review and editing

### Author ORCIDs

Sivan Henis-Korenblit https://orcid.org/0000-0001-8023-6336

### Decision letter and Author response

Decision letter https://doi.org/10.7554/eLife.50896.sa1
Author response https://doi.org/10.7554/eLife.50896.sa2

## Additional files

### Supplementary files

• Supplementary file 1. Lifespan analysis of mutants with defective processing of endo-siRNA and inactivation of *ptp-5.1*.

• Supplementary file 2. 72 genes whose levels decreased by more than 1.5 fold in *dcr-1 glp-1* double mutants compared to *glp-1* single mutants (p-value<0.05).

• Supplementary file 3. GO analysis of 72 genes whose levels decreased by more than 1.5 fold in *dcr-1 glp-1* double mutants compared to *glp-1* mutants (p-value<0.05).

• Supplementary file 4. 132 genes whose levels increased by more than 1.5 fold in *dcr-1 glp-1* double mutants compared to *glp-1* single mutants (p-value<0.05).

• Supplementary file 5. Expression of secondary siRNAs of *dcr-1 glp-1* vs. *glp-1* mutants (Padj <0.05).

• Supplementary file 6. DAVID analysis of 132 genes whose levels increased by more than 1.5 fold in *dcr-1 glp-1* double mutants compared to *glp-1* mutants (p-value<0.05).

• Supplementary file 7. RNAi lifespan screen of *dcr-1 glp-1* double mutant.

• Supplementary file 8. Statistical data.

• Transparent reporting form

### Data availability

Raw and processed high-throughput sequencing data and microarray data generated and/or analyzed during this study were deposited under the Gene Expression Omnibus, with accession number GSE122457 and GSE128935. All other data generated or analysed during this study are included in the manuscript and supporting files.

The following datasets were generated:

| Author(s) | Year | Dataset title | Dataset URL | Database and Identifier |
|---|---|---|---|---|
| Cohen BM, Ben-Hemo S, Fishman A, Waldman B-AH, Lamm AT, Henis-Korenblit S | 2018 | endo-siRNA induced inactivation of a neddylation suppresor promotes longevity and HSF-1 activation in germline-less animals | https://www.ncbi.nlm.nih.gov/geo/query/acc.cgi?acc=GSE122457 | NCBI Gene Expression Omnibus, GSE122457 |
| Berkman MC, Hemo SB, Fishman A, Waldman B-AH, Lamm AT, Henis-Korenblit S | 2019 | Endogenous siRNAs promote proteostasis and longevity in germline-less Caenorhabditis elegans | https://www.ncbi.nlm.nih.gov/geo/query/acc.cgi?acc=GSE128935 | NCBI Gene Expression Omnibus, GSE128935 |

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
