## [Decision Letter]

**Acceptance summary:**

This manuscript describes a role for endogenous small interfering RNAs (siRNA) in lifespan extension. Using long-lived germ-line ablated *C. elegans*, the authors show that the machinery required for maturation siRNA are required for enhanced proteostasis and lifespan extension. They identify a specific siRNA target, the tyrosine phosphatase ptp-5.1, that is also required for these effects, supporting the model that siRNA regulation of ptp-5.1 mediates longevity.

**Decision letter after peer review:**

Thank you for submitting your article "Endogenous siRNAs promote proteostasis and longevity in germline less *C. elegans*" for consideration by *eLife*. Your article has been reviewed by three peer reviewers, one of whom is a member of our Board of Reviewing Editors, and the evaluation has been overseen by David Ron as the Senior Editor. The reviewers have opted to remain anonymous.

The reviewers have discussed the reviews with one another and the Reviewing Editor has drafted this decision to help you prepare a revised submission.

This manuscript entitled "Endogenous siRNAs promote proteostasis and longevity in germline-less *C. elegans*" showed that endo-siRNAs are necessary for the long lifespan and the maintenance of proteostasis of glp-1 mutants, which do not have germline. They showed that endo-siRNA-regulating factors, including dicer, are required for the longevity of glp-1 mutants. They found that dicer was necessary for the activation of HSF-1, a master transcription factor for cytosolic chaperone expression, and proteostasis in the glp-1 mutants. Using transcriptomic analysis, they identified a tyrosine phosphatase, ptp-5.1, as an indirect target of endo-siRNA, whose genetic inhibition restored the longevity and the activation of HSF-1 in glp-1 dicer double mutants. These data suggest that down-regulation of ptp-5.1 by endo-siRNA promotes longevity and improves proteostasis in germline-deficient animals. The findings are intriguing and will provide useful information about the role of endo-siRNAs in aging regulation.

Essential Revisions:

1) It is possible that the low-level heat shock required to induce loss of GSCs could be contributing to the involvement of chaperones here. Because all of the experiments in this paper are done with the ts allele of glp-1, it is unclear whether the model proposed is general for GSC depletion or specific for ts glp-1. It is important to provide some evidence that these observations are general rather than specific for ts glp-1. This could be done by testing whether an alternative method for extending lifespan via depletion of GSCs, such as ablation, has similar requirements for lifespan extension.

2) One of the interesting and surprising findings in the paper is that ptp-5.1 mRNA level is extremely low in both glp-1 mutants and dcr-1 glp-1 mutants, despite the increase in the expression by dcr-1 mutation in glp-1 mutants. This small difference appears to exert a robust functional and physiological consequence in shown by using ptp-5.1 mutants (Figure 4C and Figure 5). Therefore, we think it is important to test whether they can detect ptp-5.1 expression in the soma of dcr-1 glp-1 mutant adult worms (in comparison to glp-1 mutants) by using transgenic animals and/or in situ hybridization.

3) The Western Blot in Figure 2B is of poor quality. This needs to be cleaned up with similar loading control levels. It seems obvious that HSF-1 is elevated in dcr-1 glp-1 animals and probably in the single mutants as well. The claim that there is no significant difference in dcr-1 and glp-1 single mutants does not pass the "eye test". While it appears that this experiment was done 6 times, if this is the best looking of the Westerns, that's an additional concern. Why are only comparisons between glp-1 vs WT and glp-1 vs glp-1 dcr-1 shown? It would be good to see the data from all 6 replicates.

4) Discussion section, you need to be careful about describing changes to the "shapes of lifespan curves". As far as we can tell, there is no formal analysis of mortality here, which would be necessary to statistically assess whether the shape (i.e. slope or intercept of the ln(mortality) plot) is altered. Same comment about claiming changes in maximal lifespan. What statistical test was performed to assess maximal lifespan differences?

5) There seem to be some inconsistencies between the model and the data which need to be addressed. For example, in Figure 2A, cytosolic chaperone levels are not increased in glp-1 mutants compared to wild-type but decreased by dcr-1 mutations. How can this be explained with the role of HSF-1 in the longevity of glp-1 mutants? This also does not fit the model regarding inhibition of HSF-1 by ptp-5.1, as ptp-5.1 mRNA level was drastically reduced in the glp-1 mutants, and therefore the expectation is HSF-1 up-regulation in glp-1 mutants compared to wild-type.

---

## [Author Response]

This manuscript entitled "Endogenous siRNAs promote proteostasis and longevity in germline-less *C. elegans*" showed that endo-siRNAs are necessary for the long lifespan and the maintenance of proteostasis of glp-1 mutants, which do not have germline. They showed that endo-siRNA-regulating factors, including dicer, are required for the longevity of glp-1 mutants. They found that dicer was necessary for the activation of HSF-1, a master transcription factor for cytosolic chaperone expression, and proteostasis in the glp-1 mutants. Using transcriptomic analysis, they identified a tyrosine phosphatase, ptp-5.1, as an indirect target of endo-siRNA, whose genetic inhibition restored the longevity and the activation of HSF-1 in glp-1 dicer double mutants. These data suggest that down-regulation of ptp-5.1 by endo-siRNA promotes longevity and improves proteostasis in germline-deficient animals. The findings are intriguing and will provide useful information about the role of endo-siRNAs in aging regulation.Essential Revisions:1) It is possible that the low-level heat shock required to induce loss of GSCs could be contributing to the involvement of chaperones here. Because all of the experiments in this paper are done with the ts allele of glp-1, it is unclear whether the model proposed is general for GSC depletion or specific for ts glp-1. It is important to provide some evidence that these observations are general rather than specific for ts glp-1. This could be done by testing whether an alternative method for extending lifespan via depletion of GSCs, such as ablation, has similar requirements for lifespan extension.

To avoid the temperature shift, which is required for the temperature dependent depletion of the germline by the *glp-1(ts)* mutation, we sought for alternative ways to deplete the germline, which do not involve a temperature shift. To this end, we have re-examined the requirement of the *dcr1* helicase activity for the lifespan extension conferred by preventing germline proliferation upon dietary supplementation of dihomo-γ-linolenic acid (DGLA). Previous studies have shown that this treatment interferes with germline proliferation ^1^, extends lifespan^2,3^ and confers superior proteostasis (similar to that observed in germline-less *glp-1* animals)^2^. We found that DGLA-mediated germline depletion extended the lifespan of wild-type animals, but failed to extend the lifespan of *dcr-1(mg375)* mutants. Hence, the endo-siRNA pathway appears to be required for the longevity of germline-less animals in general, and does not merely reflect a technical requirement stemming from the temperature shift of the temperature sensitive *glp-1* mutants. These experiments are described in subsection “Endo-siRNAs contribute to the longevity of germline-less animals”, Figure 1B and Supplementary file 1.

2) One of the interesting and surprising findings in the paper is that ptp-5.1 mRNA level is extremely low in both glp-1 mutants and dcr-1 glp-1 mutants, despite the increase in the expression by dcr-1 mutation in glp-1 mutants. This small difference appears to exert a robust functional and physiological consequence in shown by using ptp-5.1 mutants (Figure 4C and Figure 5). Therefore, we think it is important to test whether they can detect ptp-5.1 expression in the soma of dcr-1 glp-1 mutant adult worms (in comparison to glp-1 mutants) by using transgenic animals and/or in situ hybridization.

We have generated transgenic animals expressing an extrachromosomal array of *Pptp-*

*5.1::genomic ptp-5.1::gfp*. As expected, we did not detect expression of the transgene in most of the transgenic animals (indicating that the transgene was either not expressed or expressed below our detection threshold). Surprisingly, we did detect somatic expression of the transgene in 15% of both wild-type and GSC(-) *glp-1* animals. Specifically both the wild-type and the *glp-1* animals that expressed the reporter displayed a clear fluorescent signal specifically in two adjacent cells in the mid-intestine. Interestingly, inactivation of the endo-siRNA pathway (by introduction of a *dcr-1* mutation) doubled the fraction of the animals expressing the reporter in their mid-intestine, but this was independent of the removal of the germline. Based on this we conclude that *ptp-5.1* is expressed in 2 intestinal cells in a fraction of the animals. We hypothesize that repression of the expression in these cells is important for HSF-1 activation in *glp-1* animals. These experiments are described in subsection “*ptp-5.1* transcript levels are indirectly regulated by endo-siRNAs in GSC(-) animals”, Figure 4C and Figure 4—figure supplement 2.

3) The Western Blot in Figure 2B is of poor quality. This needs to be cleaned up with similar loading control levels. It seems obvious that HSF-1 is elevated in dcr-1 glp-1 animals and probably in the single mutants as well. The claim that there is no significant difference in dcr-1 and glp-1 single mutants does not pass the "eye test". While it appears that this experiment was done 6 times, if this is the best looking of the Westerns, that's an additional concern. Why are only comparisons between glp-1 vs WT and glp-1 vs glp-1 dcr-1 shown? It would be good to see the data from all 6 replicates.

We added an additional replicate (total of 7 replicates), re-quantitated all the data and did the statistical analysis on all 7 experiments, including comparisons between all strains. Based on the updated results, the statistical analysis indicates that HSF-1 levels significantly increase in *glp-1* mutants (in 6/7 repeats, P=0.004) and in *dcr-1 glp-1* double mutants (in 7/7 repeats, P=0.002) compared to WT. *dcr-1* mutants also had more HSF-1 upon averaging across all 7 experiments compared to WT animals (in practice a clear increase was observed in 4/7 experiments), but this was not statistically significant (P=0.0559). ANOVA test between the remaining strains (dcr-1/glp-1/dcr-1 glp-1) showed no significant difference among them (P=0.0649). This includes the comparison which interested us the most – the comparison between *glp-1* and *dcr-1 glp-1*. In addition, we replotted the data as boxplots and added representation of each individual experiment (Figure 2B). We present a western blot of an experiment with values similar to those of the combined analysis (Figure 2B) See subsection “Endo-siRNAs are required for activation of the heat shock response in GSC(-) animals”, Figure 2B, Supplementary file 8.

4) Discussion section, you need to be careful about describing changes to the "shapes of lifespan curves". As far as we can tell, there is no formal analysis of mortality here, which would be necessary to statistically assess whether the shape (i.e. slope or intercept of the ln(mortality) plot) is altered. Same comment about claiming changes in maximal lifespan. What statistical test was performed to assess maximal lifespan differences?

The parts referring to the shape of the lifespan curves and the maximal lifespan have been removed.

5) There seem to be some inconsistencies between the model and the data which need to be addressed. For example, in Figure 2A, cytosolic chaperone levels are not increased in glp-1 mutants compared to wild-type but decreased by dcr-1 mutations. How can this be explained with the role of HSF-1 in the longevity of glp-1 mutants?

Because the germline accounts for about two-thirds of all adult nuclei, one needs to be careful when comparing expression levels between animals that contain a germline to those that lack a germline. Hence, we were very careful throughout the manuscript to compare GSC(-) animals to GSC(-) animals (i.e. *glp-1* mutants and *dcr-1 glp-1* double mutants). This is an important point, and thus we now added this point to the text in the legend of Figure 2A. Taking this into account, the high expression of chaperones in WT animals compared to GSC(-) animals could be accounted for by expression of chaperones in the germline cells.

As for the effects of *dcr-1* inactivation on chaperone expression – we detected a significant reduction in the expression of all 3 chaperone transcripts examined in GSC(-) animals. In contrast, we detected a significant reduction in the expression of only one of the chaperones in the case of GSC(+) animals. Thus, this seems to be a different mode of regulation specific for this gene rather than the regulation of the upstream transcription factor HSF-1. legend of Figure 2A.

This also does not fit the model regarding inhibition of HSF-1 by ptp-5.1, as ptp-5.1 mRNA level was drastically reduced in the glp-1 mutants, and therefore the expectation is HSF-1 up-regulation in glp-1 mutants compared to wild-type.

Even though we cannot compare between chaperone levels between WT (=soma+germline) and *glp-1* animals (=soma only) (because of the presence vs. absence of the germline), we can compare the activation of HSF-1 in the soma of these strains, by looking at HSF-1 foci formation in the soma. Indeed, consistent with the predictions of our model, in this set of experiments HSF-1 formed more foci in glp-1 mutants compared to WT animals (Figure 2C-D).

Nevertheless, we extended the model to demonstrate that removal of *ptp-5.1* is required, but not sufficient for activating HSF-1. See Figure 6.